# A deep learning technique-based data-driven model for accurate and rapid flood predictions in temporal and spatial dimensions

Qianqian Zhou[1], Shuai Teng[1], Zuxiang Situ[1], Xiaoting Liao[1], Junman Feng[1], Gongfa Chen[1], Jianliang Zhang[2], Zonglei Lu[3]

[1]School of Civil and Transportation Engineering, Guangdong University of Technology, Guangzhou, 510006, China

[2]Guangdong Communication Planning & Design Institute Group Co., Ltd, Guangzhou, 510507, China

[3]GRUNDFOS Pumps (Shanghai) Co., Ltd. Guangzhou Branch, Guangzhou, 510095, China

*Correspondence to*: Qianqian Zhou (qiaz@foxmail.com)

**Abstract.** An accurate and rapid urban flood prediction model is essential to support decision-making on flood management. This study developed a deep learning technique-based data-driven model for flood predictions in both temporal and spatial dimensions, based on an integration of LSTM network, Bayesian optimization and transfer learning techniques. A case study in north China was applied to test the model performance and the results clearly showed that the model can accurately predict the maximum water depths and flood time series for various hyetograph inputs, meanwhile with substantial improvements in computation time. The model predicted flood maps 19,585 times faster than the physical-based hydrodynamic model and achieved a mean relative error of 9.5%. For retrieving the spatial patterns of water depths, the degree of similarity of the flood maps was very high. In a best case, the difference between the ground truth and model prediction was only 0.76% and the spatial distributions of inundated paths and areas were almost identical. With the adoption of transfer learning, the proposed model was well applied to a new case study and showed robust compatibility and generalization ability. Our model was further compared with two baseline prediction algorithms (ANN and CNN) to validate the model superiority. The proposed model can potentially replace and/or complement the conventional hydrodynamic model for urban flood assessment and management, particularly in applications of real time control, optimization and emergency design and plan.

## 1 Introduction

Flooding has been one of the most frequent and disturbing disasters in many urban areas, especially under impacts of climate change and urbanization (Arnone et al., 2018; Zhou et al., 2019; Kaspersen et al., 2017; Mahmoud and Gan, 2018). Prediction of flooding plays a key role in urban flood evaluation and management and can provide effective decision aid tools to reduce flooding impacts on both society and environment (Lowe et al., 2017; Xie et al., 2017; Hou et al., 2021a). Establishing rapid and accurate flood prediction methods is thus essential, however, a complicated and challenging task (Guo et al., 2021). Conventionally, hydrodynamic models have been employed for applications such as flood inundation simulations, assessment of mitigation and adaptation measures (Wolfs and Willems, 2013; Wang et al., 2021; Li et al., 2019). Despite the fact that the physically-based models can well simulate the drainage and surface flooding processes, they usually require a large number of inputs to describe the model structure and parameters, and are often computational intensive, especially with the adoption of two-dimensional calculations (Yin et al., 2020; Jamali et al., 2018; Ziliani et al., 2019; Hou et al., 2021b). Meanwhile, there is an inevitable need for conceptualization and simplification in the physical model, and the relevant calibration and validation procedures are also quite challenging (Davidsen et al., 2017; Coulthard et al., 2013; Wu et al., 2018).

Machine learning (ML) provides a potential detection tool for the above challenges. A number of scholars have explored the performance of ML-based models in urban flood mapping at an early age. Berkhahn et al. (2019) applied an artificial neural network (ANN) to predict the maximum water levels during a flash flood event and used a growth algorithm to find the suitable topology of ANN. Lin et al. (2020) tested different neural network algorithms and found that the elastic back propagation network performed the best, meanwhile with introducing clustering to preprocess the discharge curves to improve the predictions. Hou et al. (2021a) combined the random forest (RF) and K-nearest neighbour (KNN) machine learning algorithms with a hydrodynamic model to develop a fast urban flood inundation forecasting model. Although these ML algorithms have achieved relatively satisfactory results, their detection efficiencies were still poor. In particular, the generalization ability of the models was weak, which limited their applications in practical prediction tasks.

To solve the bottleneck problems on complex model construction and high computation cost of the hydrodynamic models, the potential of novel deep learning (DL) techniques in capturing and predicting

flooding processes have been increasingly explored in recent years to alleviate the burden on physical modelling (Han et al., 2021; Guo et al., 2021; Hou et al., 2021a). The DL methods harbour intelligent learning mechanisms and can extract learning data features from historical knowledge. The methods can find the relationships between input and output data with a much lower computational cost, in particular

with high-performance computers. It has been demonstrated that these DLs have excellent generalization capabilities so that even complex data features (e.g., flood pattern and tendency) could be automatically learned with a high prediction accuracy and computation efficiency (Lecun and Bengio, 1995; Rawat and Wang, 2017; Guo et al., 2021; Yosinski et al., 2014). With proper data provided, the methods can learn the flood patterns through data features and eliminate the analysis of the actual physical processes. The

high computation efficiency is essential, especially for flooding impacts modelling in urban areas with complex local conditions and high spatial resolutions.

A number of attempts on DL applications are found in the field of drainage and flood condition detection and assessment. Moy De Vitry et al. (2019) used a deep convolutional neural network (DCNN) approach for scalable flood level trend monitoring with data from surveillance camera systems. Han et al.

(2021) proposed a YOLO-based DL framework to automatically monitor the urban road inundation under dry and wet conditions. Hou et al. (2021c) performed an experimental flooding test using surveillance cameras to obtain flood images and employed a Mask R-CNN (mask region-based convolutional neural network) to detect and segment the inundated areas in river channels. Guo et al. (2021) adopted a DCNN-based approach for urban flood predictions and achieved satisfying prediction

accuracy and computation efficiency. Löwe et al. (2021) further proposed a CNN with U-Net architecture to predict urban flood hazards at high resolution and short time scales, by taking into account multiple spatial and rainfall variables as input datasets. Hofmann and Schüttrumpf (2021) introduced a DL model floodGAN to predict 2D flood inundation, and an image-to-image based translation method was used to convert flood hazard maps directly from raw rainfall images. These

studies have shown the potential of DL techniques in a wide range of flow-related problems with promising accuracy and low computational cost. Nevertheless, the previous studies have been focused on the prediction of maximum flood maps and research on time-series predictions has been lacking.

Different from other popular DL algorithms, the long short-term memory (LSTM) network allows inputs of unequal dimensions/lengths, which is especially suitable for processing time-series data, such

as traffic flow (Xia et al., 2021) and power systems (Ciechulski and Osowski, 2021). All these studies

have demonstrated the remarkable capabilities of LSTM in data feature learning in the time dimension. Zhu et al. (2020) developed a probabilistic LSTM network coupled with Gaussian process (GP) to improve the streamflow forecasting in the upper Yangtze River. Despite the advances of the studies, most of which focused on a large spatial scale and required several types of input data (e.g., rainfall, terrain,

flow depth) for model predictions. So far, no study has explored the LSTM performance on automated prediction of urban-scale flood inundation at high resolution, not to mention the combination of optimization algorithms and transfer learning methods to enhance the model performance, generalization and compatibility.

The goal of this study is to provide a novel end-to-end method for a dynamic, rapid and accurate

urban flood prediction for real-time evaluation and emergency decision-making. Given the uncertainty/unknown of rainfall events and the advantages of LSTM, we present a DL-based technique with an integration of LSTM network, Bayesian optimization and transfer learning methods. The inundation areas and water depths can be forecasted in both temporal and spatial dimensions with only rainfall inputs. Both the maximum water depths and flood time series can be predicted. The method was

firstly tested in a case study in northern China with various rainfall conditions. Another case study was used to test the compatibility and generalization ability of the developed model. Finally, two classical flood prediction algorithms (ANN and CNN) were considered as the baseline models to confirm the effectiveness of the proposed method.

## 2 Methodology and data

To examine the performance of the proposed approach, we firstly selected two case studies and obtained the relevant data describing the rainfall inputs, local topography and drainage systems. A coupled 1D-2D hydrodynamic model was employed to simulate the inundation areas and water depths at different time steps under various design rainfall events. Then the DL technique-based prediction model was established and trained based on the simulated flood maps and tested with random rainfall inputs to

examine the relevant prediction accuracy and computation cost. The Bayesian optimization and transfer learning techniques were adopted to enhance the detection performance and generalization ability of the developed model.

**2.1 Case study areas**

A portion of the city Hohhot, the capital of the Inner Mongolia Autonomous Region, was used as the case
study to test the performance of proposed method. The city is located in Northern China and within a cold
semi-arid climate zone. The winters are dry but the summers can be very hot and rainy. The average
annual rainfall was approximately 396 mm, with majority of which concentrated from July to August
(Zhou et al., 2018; Zhou et al., 2016). The detailed landuse (Fig. 1a) mainly consists of residential areas,
commercial districts, institutes, green spaces and other landuse. The terrain is high in the north and lower
in the south (Fig. 1b) and thus the runoffs generally flow in a north to south direction. The service level of
the drainage system was rather low and the original design return period of the system was below once a
year (Zhou et al., 2018). In recent years, the flooding has occurred more frequently in the area.
Nevertheless, there is a lack of accurate historical data on flood areas and depths and thus simulations of
flood events were performed with a 1D/2D coupled hydrodynamic model (to be introduced in the
following sections) under various design rainfalls.

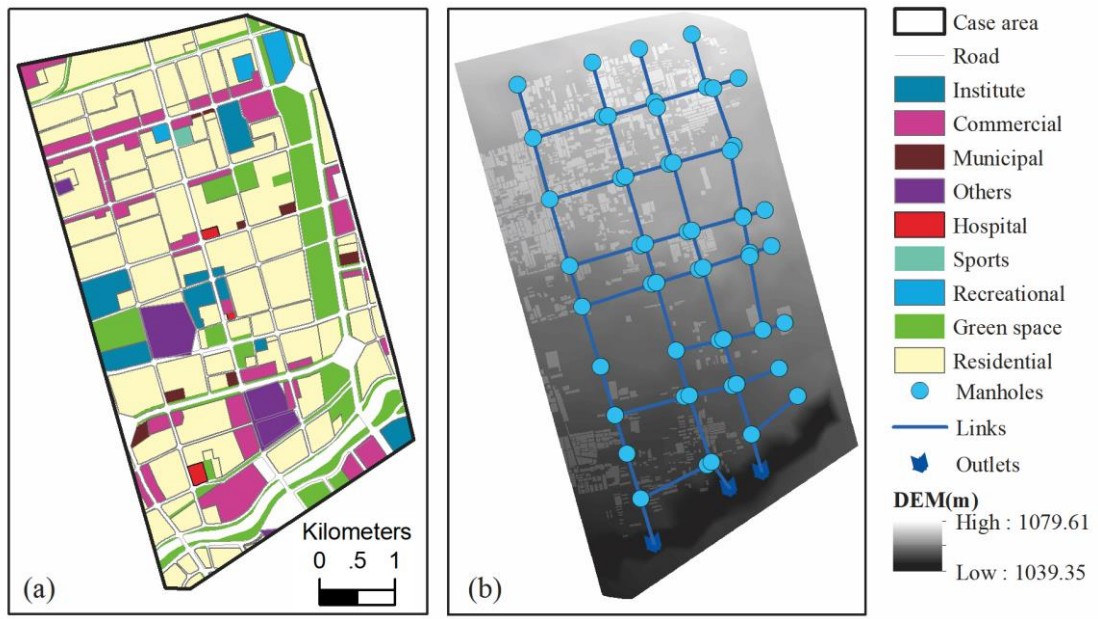

**Figure 1: case study (a) landuse and (b) drainage system with DEM descriptions.**

The input rainfall hydrographs for model training and validation were calculated using the regional
Storm Intensity Formula (SIF) ($q=635\times(1+0.841\times\lg(P))/t^{0.61}$, where q is the storm intensity
(($L/s)/hm^2$), p is the design return period (a) and t is the rainfall duration (minutes), respectively) (Zhang
and Guan, 2012; Zhou et al., 2016). The rainfall calculation followed the national code for design of
outdoor drainage (Mohurd, 2016) and the design principles of Chicago Design Storms (Berggren et al.,

2014; Panthou et al., 2014; Zhou et al., 2012). The detailed procedures in applying the regional SIF to obtain CDSs are outlined in the national Technical Guidelines for Establishment of

Intensity-Duration-Frequency Curve and Design Rainstorm Profile (Mohurd, 2014).

In this case study, we investigated the DL-based model ability in predicting both maximum water depth and flood time series during the entire rainfall. Two types of datasets were established: (1) maxH dataset, i.e., the maximum flood depth: there were in total 90 rainfall events adopted, with return periods ranging from 1 to 100 years and rainfall duration of 2, 4 or 6 hours, respectively. Meanwhile, three types

of rain peak position coefficients were tested, including 0.3, 0.5 and 0.7. All rainfall inputs were generated with a temporal resolution of 10 minutes. The details on the return period, rainfall duration and peak position of the input rainfall events are provided in the supplementary material SF 1. Among the 90 flood maps, 90% of which were randomly selected for model training and validation, and the rest 10% for testing. (2) time series dataset: we adopted 11 rainfall events and recorded the flooded water

depths every 10 minutes for the entire case study under each rainfall. Among that, nine rainfalls were used for training and the other two rains for testing. Furthermore, a second study was adopted to test the capability of the developed model to generalize to different case studies or contexts. The area is in a relatively distant location in the Hohhot city and Fig. 2 shows the main landuse, drainage system and surface elevation of the case study. In this case study, the DL-based model was mainly tested on the

basis of maxH predictions.

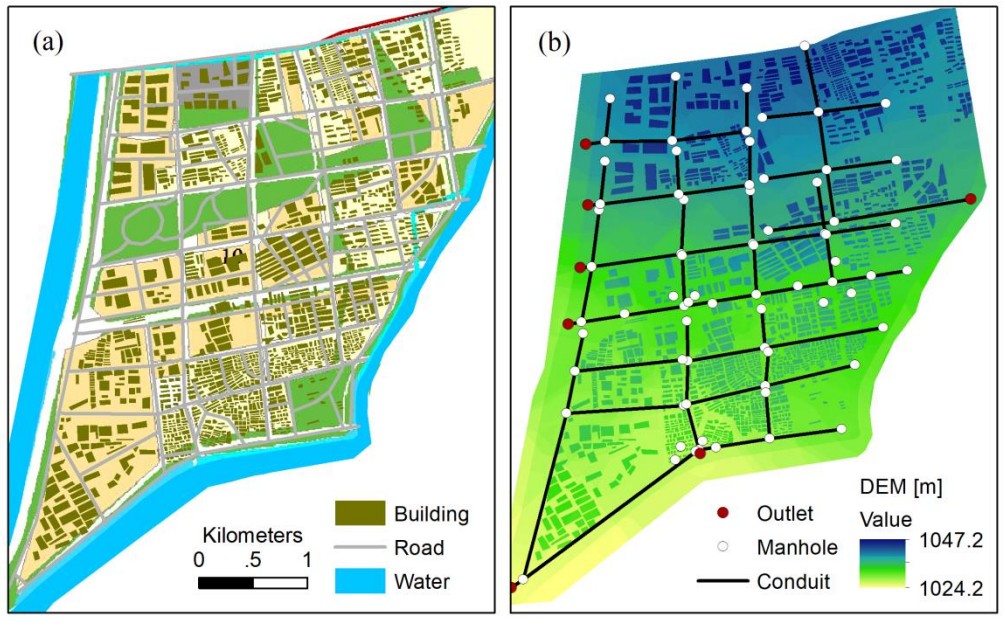

**Figure 2: (a) landuse and (b) drainage system and DEM of the second case study.**

**2.2 Physically-based hydrodynamic model**

All the overland flooding maps and time series were simulated using the 1D-2D coupled hydrodynamic model, namely Mike Urban (Mike by Dhi, 2016). The hydrological inputs and pipe flows were simulated using the 1D Mouse model and the overland flows were calculated using the Mike 21 model. With the precipitation inputs, the runoff model was characterized by the general catchment data, such as locations, areas, imperviousness, and time of concentration. The shape of the runoff hydrograph was computed by the 'Time-Area' method (Mike by Dhi, 2016). The calculation of unsteady flow in the pipe network was conducted by solving the Saint Venant equations, which are the vertically integrated equations of conservation of continuity (i.e., continuity equation) and momentum (i.e., momentum equation). The surface inundation model was established by the MIKE 21 rectangular grid component and the links between the 1D and 2D models were established to simulate the flow interactions between the pipeline and overland flows. Especially, the flow in the links was governed by an orifice equation (Mike by Dhi, 2016). When the underground drainage was surcharged, the excess water flowed to the surface and conducted surface inundation calculations under the context of extreme precipitations. On the surface, the water typically flowed along buildings or streets based on a description of the local digital elevation/topography (Mark et al., 2004; Leandro et al., 2009).

The model outputs include the overland flow paths, extents, depths and velocities at different time steps. One of the most commonly used outputs is the flood maps describing the maximum water depths caused by the given rainfall inputs (Kaspersen et al., 2017; Mike by Dhi, 2016; Zhou et al., 2012). These flood maps can be further integrated with vulnerability data for an assessment of flood risk levels at different spatial scales (Sampson et al., 2014; De Moel et al., 2009; Ashley et al., 2007). In doing so, the critical areas with higher levels of flood risks can be identified and allocated with priorities in mitigation and adaptation plans (De Moel et al., 2015; Zhou et al., 2012). Take the first case study as an example, as shown in Fig. 3 that changes in input rainfalls lead to variations in simulated flood maps. Increases in flood extents and depths are seen with rainfalls of larger return periods. Note that this study was aimed to develop a flood prediction model that is applicable to various types of case studies, not just as a surrogate model of the physical model. The physical model was used to provide training samples for the DL model to learn the flood feature extraction ability. That means, all the tested hyetographs were the inputs and all the simulated flood maps were the ground truth (GT) data to train the model network. After the training, the randomly sampled 10% flood maps were used to test the model prediction performances.

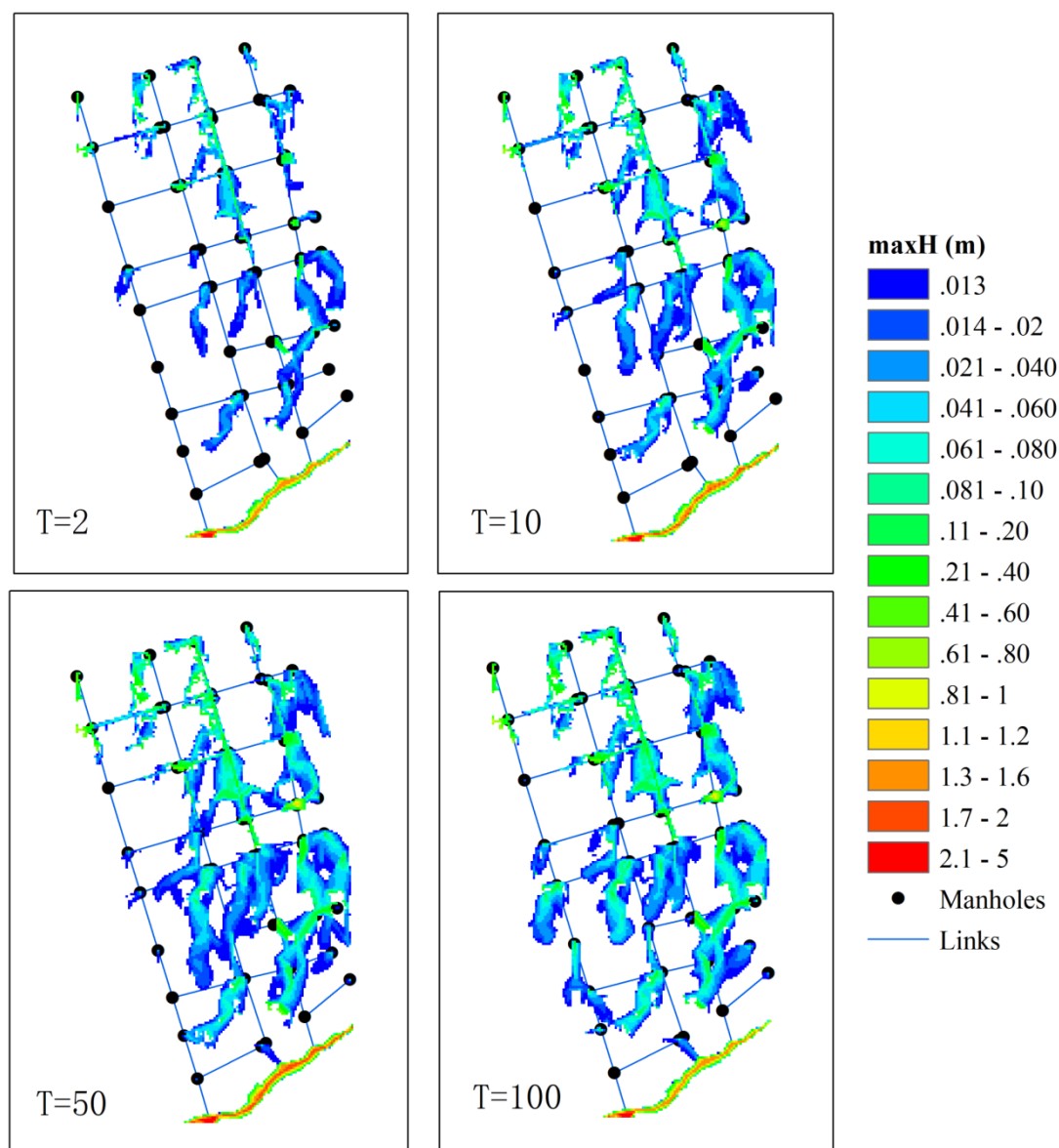

**Figure 3: Maximum flood maps simulated using the hydrodynamic model for four types of return periods.**
**The unit of T is year.**

## 2.3 Deep learning model for flood prediction

### 2.3.1 LSTM (Long Short Term Memory) network

The LSTM network has advantages in processing time-series data, especially for the long-term time-varying data. As shown in Fig. 4 that the LSTM network is used to predict the flood maps. Ideally,
the network can predict the flood depth distributions in the region as close as the real values/ ground truth (GT) values. The relative error between the output and GT values is calculated and used as a priori condition for the Bayesian optimization (BO). It is a priori probability that is used as the basis for selecting the network structure and hyper-parameter combinations in the next iteration. Finally, an

optimal network model (e.g., with an appropriate number of layers) and the corresponding

hyper-parameters are obtained through the iterative BO process.

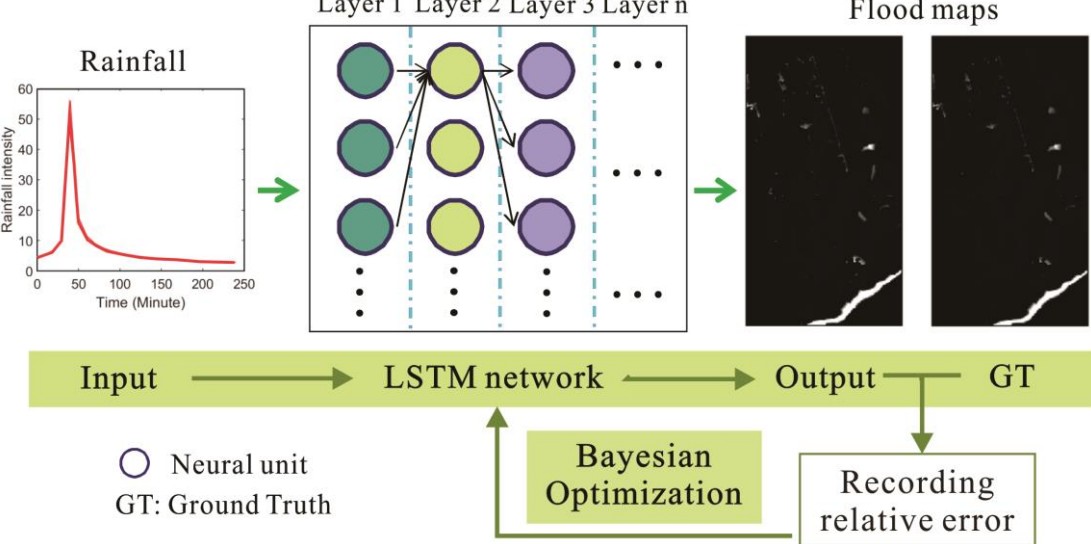

**Figure 4. Method implementation steps.**

The LSTM network requires only rainfall intensity as the input and the output is the water depth of all grid points all at once. That means LSTM outputs a 2D map directly, which describes the water

depth of the entire site. A regression algorithm is used for the LSTM model. Specifically, the input rainfall intensity is processed through multiple LSTM layers and activation layers, and finally, a regression layer outputs the water depth of all grid points. In other words, the process is akin to a fitting process, in which different rainfall intensities are matched nonlinearly to the water depth of grid points. The number of output grids can be set during LSTM modelling so that the output grid can be specified

for different sites.

It is noted that terrain feature (e.g., DEM) is a key factor in implementing flood prediction tasks. Besides, catchment hydrological properties (e.g., land use, area and imperviousness) and network parameters (pipeline distribution and capacity) also have essential impacts on flood conditions. When designing the method, we considered the impacts of all influencing factors in the network training, but

without specifying/demanding them in the model input. There are two main reasons: (1) The site information contains a vast amount of data. Although computer technology has made significant progress, DL algorithms still face significant challenges in processing high-dimensional data. Such data can significantly decrease the prediction efficiency, particularly when applied to large-scale areas. The method proposed mitigates the impacts of high-dimensional site feature data on prediction efficiency; (2)

Neural network technology involves learning and creating a mapping relationship between input and output data. The function of the LSTM model is to establish the nonlinear mapping relationships between the input rainfall intensity, the implicit influence conditions and the output flood depth to reflect the impact of these factors.

A benchmark LSTM network structure is shown in Fig. 5. With the input data (rainfall intensity), the LSTM gets the output (water depth) through a series of functional layers, including a LSTM layer (containing $N$ neural units), a Leaky ReLU activation function (Eq. (1)), and a fully connection (FC) layer. In the LSTM layer, the rainfall is input to the $N$ neural units to obtain the $N$ outputs (i.e., $h_0$, $h_1$, $h_2$, …, $h_N$). The outputs of these neural units are then transformed nonlinearly by the Leaky ReLU activation function and enter the FC layer. Eventually, the FC layer delivers the output of the network. Specifically, the LSTM network is trained through the adaptive moment estimation (Adam) optimizer (Eqs. (2-4)). Meanwhile, two performance indicators are used to reflect the network training, which are the loss (Eq. (5)) and root mean square error (RMSE, Eq. (6)) respectively.

$$f(x) = \begin{cases} x, & x \geq 0 \\ scale \times x, & x < 0 \end{cases} \tag{1}$$

$$m_l = \beta_1 m_l + (1 - \beta_1)\nabla E(\theta_l) \tag{2}$$

$$v_l = \beta_2 v_{l-1} + (1 - \beta_2)[\nabla E(\theta_l)]^2 \tag{3}$$

$$\theta_{l+1} = \theta_l + \frac{\alpha m_l}{\sqrt{v_l + \varepsilon}} \tag{4}$$

$$Loss = \frac{1}{n}\sum_{i=1}^{n}(y_i - \hat{y}_i) \tag{5}$$

$$RMSE = \sqrt{\frac{1}{n}\sum_{i=1}^{n}(y_i - \hat{y}_i)} \tag{6}$$

Where, $x$ and $scale$ are input and scale factor (0.01), receptively. Any input value that is less than zero is multiplied by a fixed scale factor. $\beta_1$ and $\beta_2$ are the Gradient Decay Factor (0.9) and Squared Gradient Decay Factor (0.999), respectively. $E(\theta)$ is the loss function, $m$ and $v$ are the momentum terms, and $\varepsilon=10^{-8}$. $n$ is the number of samples, $y_i$ and $\hat{y}_i$ are the predicted and real results, respectively.

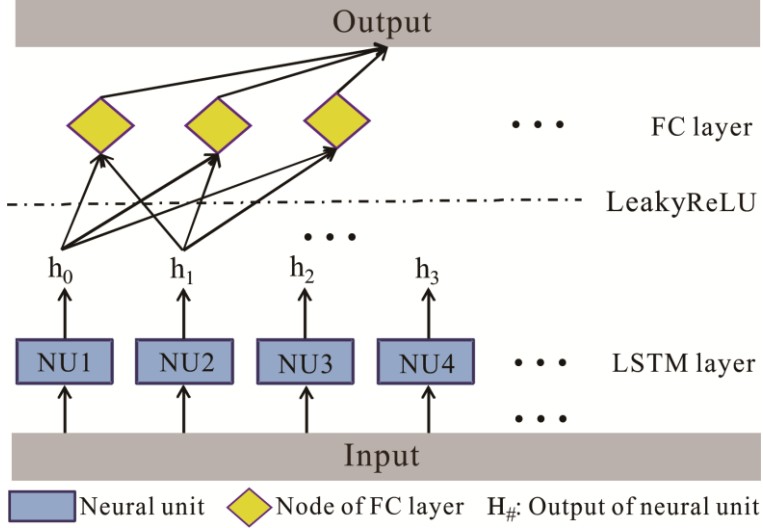

**Figure 5: Framework of the LSTM network.**

The neural unit is a key component of the LSTM network and the structure of a single neural unit is shown in Fig. 6, including a forget gate (Eq. (7)), an input gate (Eqs. (8-10)) and an output gate (Eqs. (11-12)). The forget gate determines how many unit states at time (t-1) are retained until time (t). The input gate determines the update of the unit states. The output of the LSTM neural unit state is determined by the nonlinear activation function (Sigmoid, in Eq. (13)) and the output gate. In general, an input ($x$) passes through a neural unit to get an output ($h$). Specifically, the calculation process of a single LSTM neural unit is shown as follows:

$$f_t = \sigma(\boldsymbol{W}_f[h_{t-1}, x_t] + b_f) = \sigma(\boldsymbol{W}_{fh}h_{t-1} + \boldsymbol{W}_{fx}x_t + b_f) \tag{7}$$

$$i_t = \sigma(\boldsymbol{W}_i[h_{t-1}, x_t] + b_i) \tag{8}$$

$$c'_t = \tanh(\boldsymbol{W}_c[h_{t-1}, x_t] + b_c) \tag{9}$$

$$c_t = f_t c_{t-1} + i_t c'_t \tag{10}$$

$$o_t = \sigma(\boldsymbol{W}_o[h_{t-1}, x_t] + b_o) \tag{11}$$

$$h_t = o_t \times \tanh(c_t) \tag{12}$$

$$\sigma(x) = \frac{1}{1 + e^{-x}} \tag{13}$$

Where, $f_t$ is the output of the forget gate, $\boldsymbol{W}_f$ and $b_f$ are the weight matrix and bias of the forget gate, and $h_{t-1}$ and $x_t$ are the output of the previous neural unit (time (t-1)) and the current input (time (t)), respectively. $i_t$ is the output of the input gate, $c'_t$ and $c_t$ are the unit state of the current input and current time, respectively. $o_t$ is the output of the output gate, $h_t$ is the neural unit output of time (t).

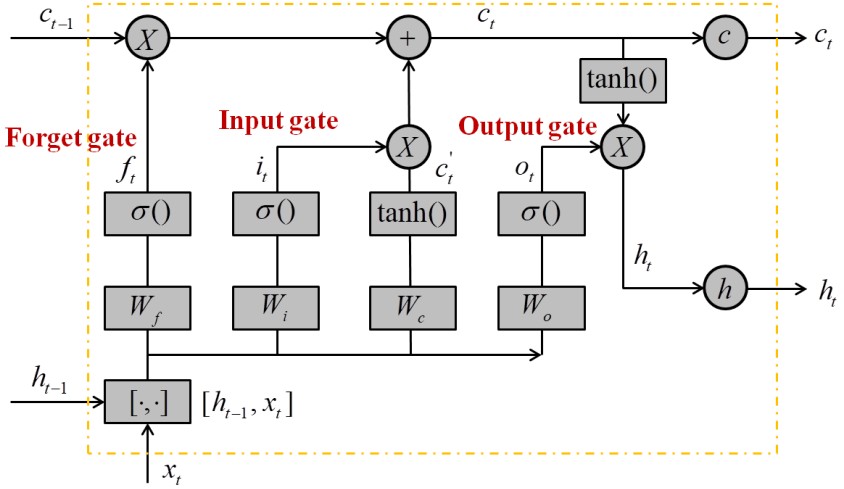


**Figure 6: The structure of single neural unit.**

In this paper, the LSTM network was built in the MATLAB 2021a (MathWorks Inc, Natick, MA, US). The input was the rainfall intensity varying with time, and the output was the maxH flood maps or water depths at different time steps of the case study sites. The training platform was performed on a
computer with NVIDIA RTX GeForce 2060 GPU, Intel Core i7-4790 @ 3.60 GHz CPU, Windows 10.

Furthermore, to validate the performance of the developed LSTM model, two additional baseline models were adopted for comparison, namely the Artificial Neural Network (ANN) and the Convolutional Neural Network (CNN). The ANN is a fully connected network (Back-Propagation neural network, BPNN) that included one input layer, one hidden layer (included 170 connection nodes/neurons)
and one output layer. The ANN network has played an important role in the early research of artificial intelligence (Sudheer et al., 2002) as it has performed well in certain simple regression tasks. However, the fully connected structure has significantly increased the computing cost of the network and limited its further applications in the big data field. On the other hand, the CNN network adopted included two convolution layers, one pooling layer, two activation layers and a fully connection layer. The CNN has
significantly reduced the network computing cost through the weight sharing and sparse connection. It has strong a feature extraction capability (Hinton and Salakhutdinov, 2006) and showed stronger performance than the BPNN in related research (Teng et al., 2022). In this paper, we compared the classic and popular ANN and CNN as baseline networks with the developed LSTM to clarify the effectiveness and novelty of the method proposed.

 **2.3.2 Bayesian optimization**

One problem with the aforementioned LSTM network is that its structure layers, learning rate, number of training epochs, mini-BatchSize and number of neural units were all unknown. To start from scratch, it can be very difficult and time consuming to manually select and fine-tune these hyper-parameters. Bayesian optimization (BO) is an algorithm that can automatically search for the optimal hyper-parameter combinations. The BO is a continuously updated probability model (Eq. (14)) and assumes that the probability of occurrence of Event A under the a priori condition of Event B is directly proportional to the probability of occurrence of the a posteriori condition of Event B. That is, for successively occurring events, the latter events are related to all previous events. It is a potential hyper-parametric optimization scheme, which means the most likely parametric combination is inferred through a number of a priori attempts (i.e., training network models with different structures).

The posterior probability of the optimization function is updated through a number of evaluations of objective function to obtain the optimal parameter combination. It can provide reference for the subsequent tried models according to the a priori conditions (i.e., historical evaluation records, which are the mean relative errors of the tried network model in this paper). When selecting the next group of parameter combinations, the algorithm made full use of the previous evaluation information to reduce the search time of the parameters. Specifically, we designed a variety of search ranges of the hyper-parameters and BO algorithm automatically took the values from the search ranges and constantly tried the network models with different structures, and then recorded the errors. In this paper, the hyper-parameters to be optimized included the number of LSTM layer, learning rate, Epoch, mini-BatchSize, and number of hidden units. The search ranges of these five parameters were set to [1-5], [$10^{-4}$-1], [0-600], [0-100] and [0-100], respectively. Finally, BO inferred the possible optimal network combination according to the historical error information. The selection process is shown in Eq. (15).

$$P(A|B) \propto P(B|A)P(A) \tag{14}$$

$$x^* = \arg\min_{x \in \chi} f(x) \tag{15}$$

Where, $P(A|B)$ and $P(A)$ are the posterior and prior probabilities of Event A, respectively, and $P(B|A)$ is the observation point probability obtained from the previous events. $f(x)$ is the objective function (i.e., the mean relative error (Eq. (16)), see the next section), $x^*$ is the optimal parametric combination, and $\chi$ is the value range of parameters.

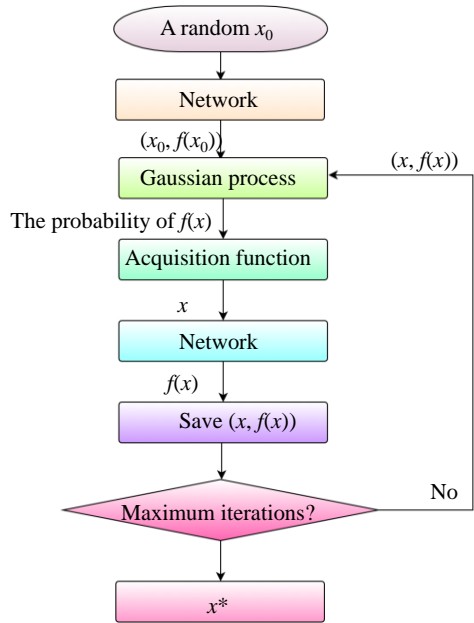

**Figure 7: The Bayesian optimization work flow.**

Specifically, the optimization work flow with regard to deep learning model is implemented following Fig. 7 : (1) A group of hyper-parameter combination $x_0$ (e.g., maxEpochs, learning rates) is randomly selected within the value ranges of the hyper-parameters; (2) The $x_0$ is input to the network for training to obtain the corresponding objective function $f(x_0)$; (3) The probability distribution of $f(x)$ corresponding to $x$ is calculated and predicted through the Gaussian process using all the inputs $(x, f(x))$; (4) The optimal $x$ is determined by the acquisition function in the probability distribution; (5) The $x$ obtained from Step (4) is taken as the hyper-parameter combination of the network to train, and calculate the objective function $f(x)$; (6) Before reaching the maximum iteration number, the $(x, f(x))$ obtained in Step (5) is used as the input of the Gaussian process to continuously update the probability model to obtain a new $(x, f(x))$. Once the maximum iteration number is reached, the $x$ corresponding to the minimum value of $f(x)$ is taken as the optimal hyper-parameter combination $x^*$.

### 2.3.3 Transfer learning (TL)

One of the main challenges of data-driven models is their compatibility, as it appears that in this study the model established is only applicable to the investigated site. The problem can be solved by using the transfer learning (TL) technology to implement flood prediction for new sites. The TL, namely learn from experience, can significantly improve the application field of intelligent algorithms. The TL is a DL method to transfer the knowledge from one domain (source domain) to another domain (target

domain), see Fig. 8. Through the training of a source model (pre-trained network) using the source data (Site A), the pre-trained network can gain a strong ability of feature extraction in the similar tasks. Subsequently, with fine-tuning (transfer learning) the new data (Site B), the pre-trained network can

quickly adapt to the new site under different scenarios. With this method, a lot of training time can be saved for the target domain (the new site), and better training effects can be achieved, especially when there are limited training samples in the target domain. In this paper, we used TL to transfer the LSTM network obtained from the current site (Site A) to the second case study site (Site B) with data from the new site, so as to expand the compatibility and generalization ability of the proposed method.

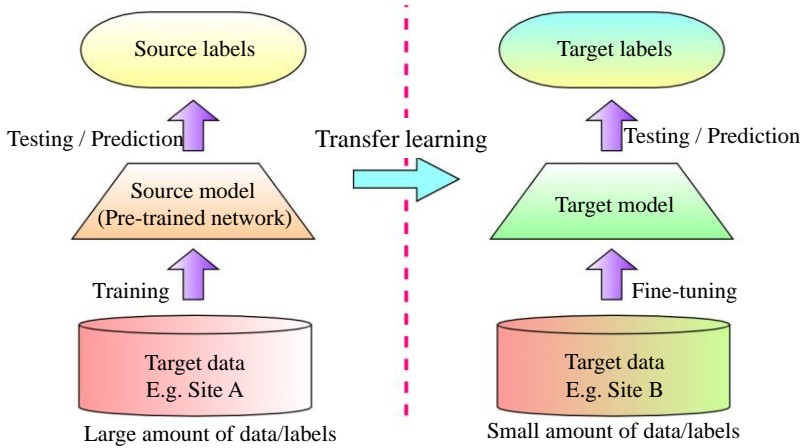


**Figure 8: Transfer learning technology.**

### 2.3.4 Performance indicators

In order to evaluate the reliability of the proposed method, five indicators were employed to evaluate the prediction results, focusing on estimating the differences in flood depths and the spatial patterns of the

flood distributions. First of all, the mean relative error (Mre) was used to calculate the depth error between the prediction results (PR) and the ground truths (GT). Next, the 2-D correlation coefficient (2D-CC) and structural similarity (SS) were used to evaluate the correlation and similarity of images (distributions of flood areas), respectively. The Bhattacharyya distance (BD) and Histogram Intersection Distance (HID) measure the similarity of two discrete or continuous probability distributions. They were

adopted to evaluate the amount of overlap between two statistical samples or images (i.e., flood maps).

$$Mre = \sum \frac{|(PR - GT)|}{GT} \tag{16}$$

$$2D - CC(I,J) = \frac{\sum_m \sum_n (I_{mn} - \bar{I})(J_{mn} - \bar{J})}{\sqrt{(\sum_m \sum_n (I_{mn} - \bar{I})^2)(\sum_m \sum_n (J_{mn} - \bar{J})^2)}} \tag{17}$$

$$SS(I,J) = \frac{(2\mu_I\mu_J + C_1)(2\sigma_{IJ} + C_2)}{(\mu_I^2 + \mu_J^2 + C_1)(\sigma_I^2 + \sigma_J^2 + C_2)} \tag{18}$$

$$BD(I,J) = -\ln\left(\sum_{x \in X} \sqrt{p(x)q(x)}\right) \tag{19}$$

$$HID(I,J) = \frac{\sum_{x \in X} \min(p(x), q(x))}{\sum_{x \in X} p(x)} \tag{20}$$

Where, $\overline{I}$ and $\overline{J}$ are the average pixel values of Image $I$ and $J$, respectively, $\mu_x$, $\mu_y$, $\sigma_x$ $\sigma_y$ and $\sigma_{xy}$ are the pixel local mean, standard deviation and cross covariance of Image $I$ and $J$, respectively. $C_1$ and $C_2$ were 6.5 and 58.5 respectively. $p(x)$ and $q(x)$ are probability distributions of pixels of Image $I$ and Image $J$, respectively. $X$ is the domain of $p(x)$ and $q(x)$.

## 3 Results and discussion

An illustration of the mean relative error of the testing dataset obtained from the 170 Bayesian optimizations is shown in Fig. 9a. The range of mean error is between 0.095 and 44.13 and the size and color of the bubble chart represents value of the mean error. It is clear that the error gradually decreased along with the iteration thanks to the optimization process. Especially, one of the networks, with a mean relative error value of 0.095, worked best in learning the flood map features. Figure 9b shows the RMSE and loss of the model with the best performance identified from the Bayesian optimization. It is shown that the loss curve stably decreased along the network training and the model achieved a convergence status after the 100 iterations with a small loss value. This implies that the DL network is very robust and trained well with the input data.

We further analyzed the influence of network parameters on the prediction results (Fig. 10). The results show that: (1) There were large errors when the values of MaxEpochs (i.e., maximum number of epochs) were set too low. Increasing the number of training epochs could avoid adverse events. (2) The MiniBatchSize had little influence on the prediction results, but it was not appropriate to take too large or small values. In this case, the MiniBatchSize of 20-70 could ensure an ideal prediction effect. (3) It is recommended to set a low learning rate. When the value was low, the achieved error was small and close to 0. (4) A deeper network layer could obtain a smaller prediction error. With the parameterization analysis, the best design scheme (network structure and hyper-parameters) of the LSTM can be determined through the Bayesian optimization. The detailed network structure is shown in Fig. 11. The learning rate, Epoch, mini-BatchSize, and number of hidden units were 0.0146, 385, 59 and 94 respectively.

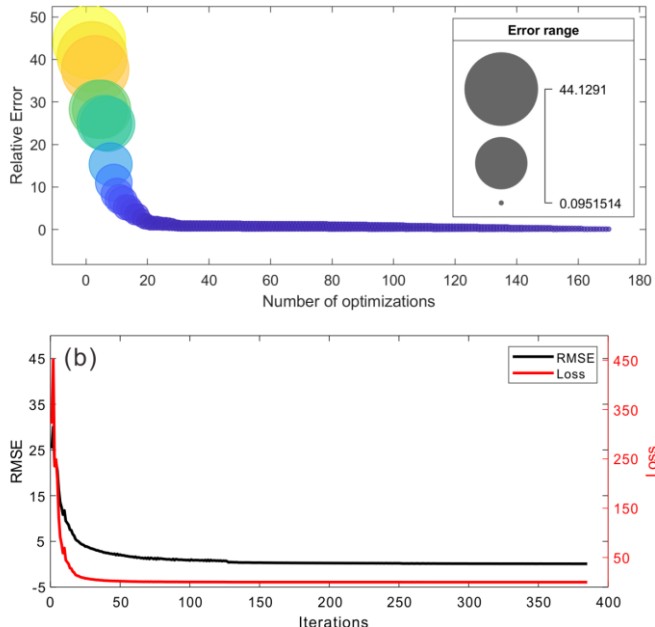

**Figure 9: (a) mean relative errors along with the Bayesian optimization process, and (b) the RMSE and loss achieved by the model with the best performance.**

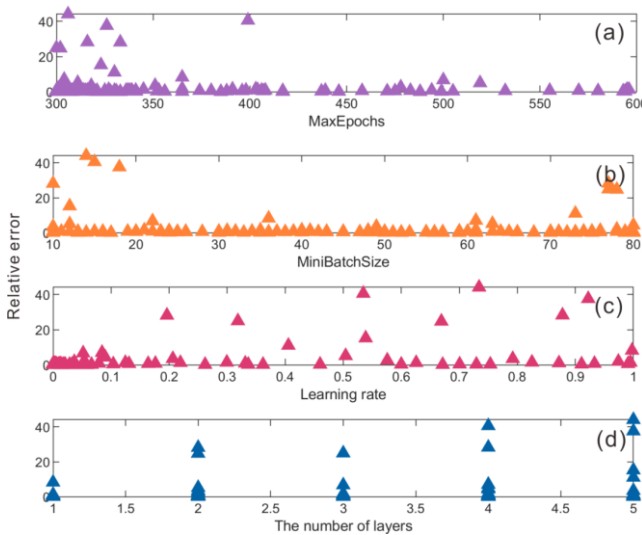

**Figure 10: Influence of four types of network parameters on model prediction performance.**

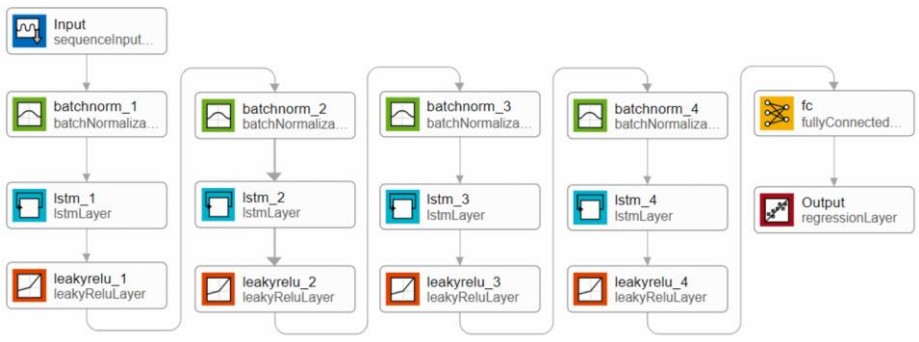

**Figure 11: The optimized model structure of the LSTM network. Batchnorm_#: Batch normalization layer, normalize the network training data (mapping raw data to [0, 1]) to speed up the training speed.**

The statistics of the performance indicators of the best performing model is analyzed in Fig. 12.

First of all, the specific value of relative error of each testing flood map was summarized in the boxplot in Fig. 12a. As reported previously, the LSTM model obtained satisfying results with a mean relative error of 9.5%. Among that, the achieved minimum RE of a single prediction was only 0.76%, which implies the predicted flood map (both the inundation locations and depths) was very close to the ground truth map for validation. The degree of similarity is illustrated by the four types of indicators in Fig. 12b.

The Bhattacharyya distances of the testing dataset were all close to zero, which meant that the spatial distributions of the ground truth and predicted flood hazard maps were very similar and a majority of the two map populations were overlapped. The ideal results were further validated by the Histogram intersection distance, structural similarity and 2D correlation coefficient as their values were all close to one. This implies that the spatial similarity of the predicted maps was very high. On the whole, the model

was proved to be superior in learning and predicting the flood maps with different hyetographs.

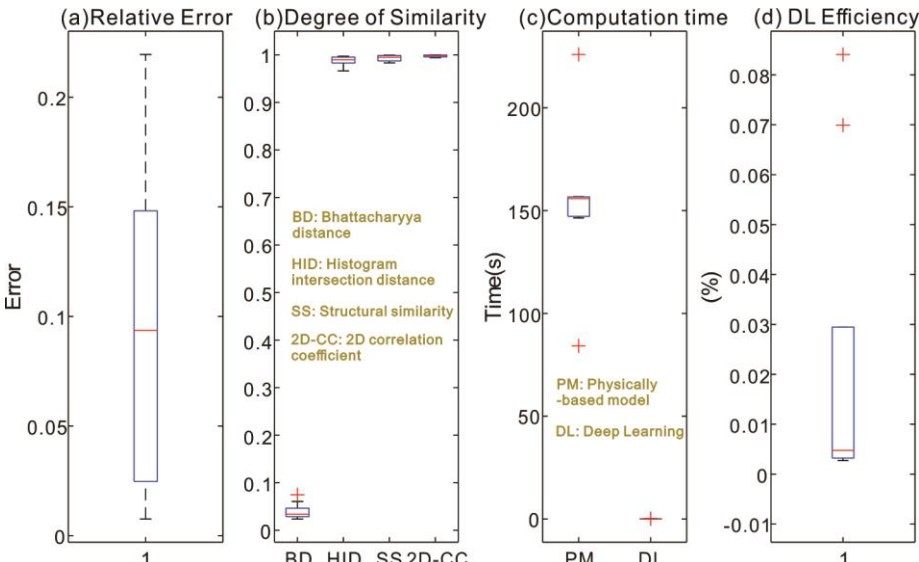

**Figure 12: (a) relative error, (b) degree of similarity, (c) computation time and (d) deep learning (DL) efficiency (i.e., computation time of DL model divided by computation time of hydrodynamic model) achieved with the testing datasets.**

The computation times of the hydrodynamic model and the DL model are compared in Fig. 12c. The average computation time of the hydrodynamic model was 153.2s, while the mean time of the prediction model was significantly reduced, with a value of 0.038s. It is shown in Fig. 12d that the hydrodynamic model took almost 19,585 times (i.e., mean value) the simulation time of the DL model. In the worst case, the hydrodynamic model simulated the flood map more than 36,600 times slower. Note

that in fact, the computation time of the hydrodynamic model was even longer, as the model needed to

run the hydrological and pipe-network+2D simulations separately and a manual integration of the two simulations were not taken into account. The results showed that with proper model training, the LSTM model was accurate and much more computational efficient, which can provide important support tools for real-time/rapid forecasting of urban flooding and emergency decision-making.

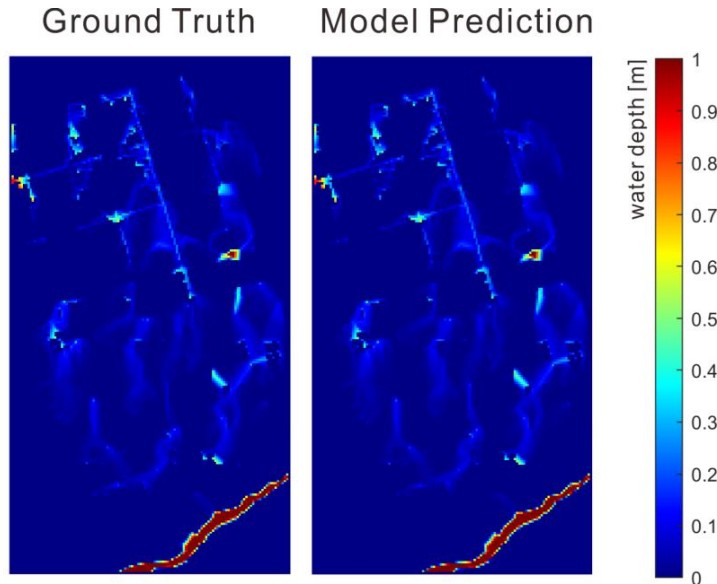

**Figure 13: Sample comparison of flood maps between ground truth and model prediction in the best case scenario.**

In visual quality, Fig. 13 illustrated the inundated areas of the ground truth and the predicted flood maps with the best model performance (i.e., with the minimum relative error). In total there were 27,183 grids in each flood map. It is seen that the LSTM model successfully retrieved the depths and spatial patterns of the inundated areas. The two maps were almost identical and it was very difficult to tell the difference without looking into further statistic details. Fig. 14a shows the spatial distributions of the relative errors of the best performing map. The differences between the two maps were almost negligible except the small regions near the water bodies. The predicted flood map could identify all the flow paths and local depressions in the ground truth map. Moreover, the spatial distributions of the mean and maximum relative errors of the testing dataset are shown in Fig. 14b and 14c. Statistics (Fig. 14d) showed that in all cases, the mean values were below 1%, indicating a good agreement between the series of predictions and the ground truth maps. The errors were much higher in the worst case, where there were a small number of cells associated with relative errors greater than 20%. Generally the errors were greater where there were higher water depths and more flow volumes. Therefore, the high-error cells were mainly located in/near the water bodies.

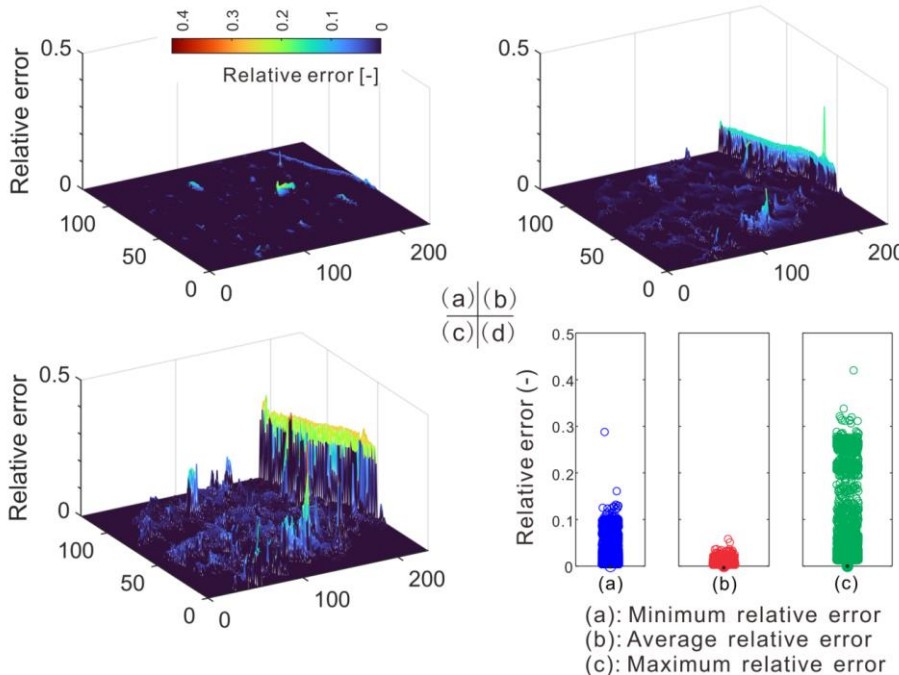

**Figure 14: Relative error of (a) best (i.e., minimum), (b) mean, and (c) worst (i.e., maximum) case scenarios, respectively. (d) summarizes the relative error data in boxplots for the three types of scenarios.**

The prediction accuracies of the deep learning model were further examined as a function of water depths in Fig. 15. Results show that the flood map dataset was imbalanced as a majority of the results contain no and shallow water. Results show that for water depth below 3m, the model performed well and most errors were below 2%. The errors tended to increase under extreme conditions, with water depths above 3.5m. Fig. 15b shows that the predicted water depths are basically consistent with the

ground truth water depths. These results clearly indicated that the deep learning model generalized well with the different hyetograph variations and could produce very accurate flood results even with only rainfall inputs.

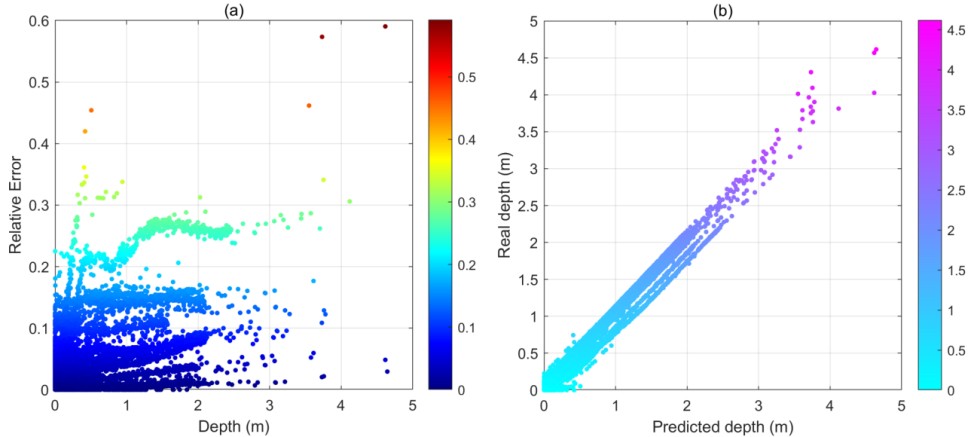

**Figure 15: (a) Relative errors of predicted flood maps as a function of water depths, and (b) ground truth**
**water depths as a function of predicted water depths.**

Fig. 16 shows the sample comparison between ground truths and model predictions of flood maps in the time dimension. It was clear that our model could well predict the flood variations at different time steps. In visual quality, the predicted flood maps were in a good agreement with ground truths at all time steps. The overall prediction effects (based on the relative error) and the evaluation indicators on the degree of similarity are summarized in Fig. 17a and Fig. 17b for the time series predictions. Larger errors may occur in the early stage of rainfall, which could be due to the impacts of drainage system on urban floods. Nevertheless, all the indicators further validated the model performance, which was also satisfying in predicting flood maps in the time dimension.

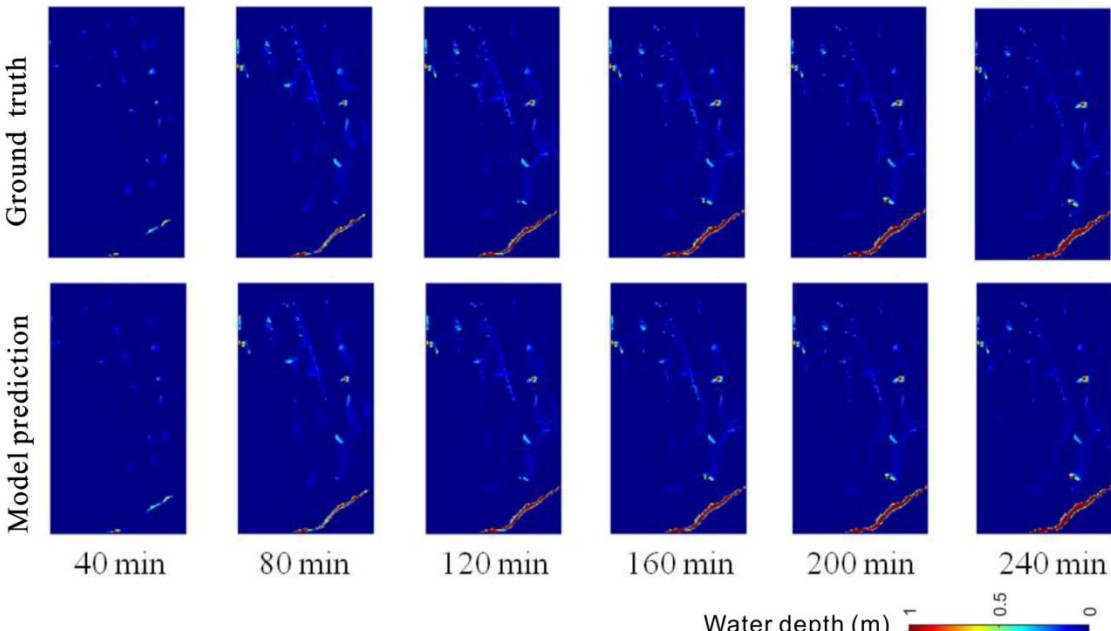

**Figure 16: Sample comparison of flood maps between ground truths and model predictions at different time steps in the first case study.**

Fig. 18 tested the performance of the established LSTM in the second case study. Results showed that with transfer leaning the proposed model was applicable and generalisable to other cases and the predicted flood maps were consistent and similar to the ground truths. Specifically, Table 1 shows the achieved performance indicators of all tested rainfall events. The obtained BD was close to 0, and HID, SS and 2D-CC were close to 1, which meant the model predictions were highly similar to the ground truth results. This proved that the flood prediction of the new site could be realized through the transfer learning technology.

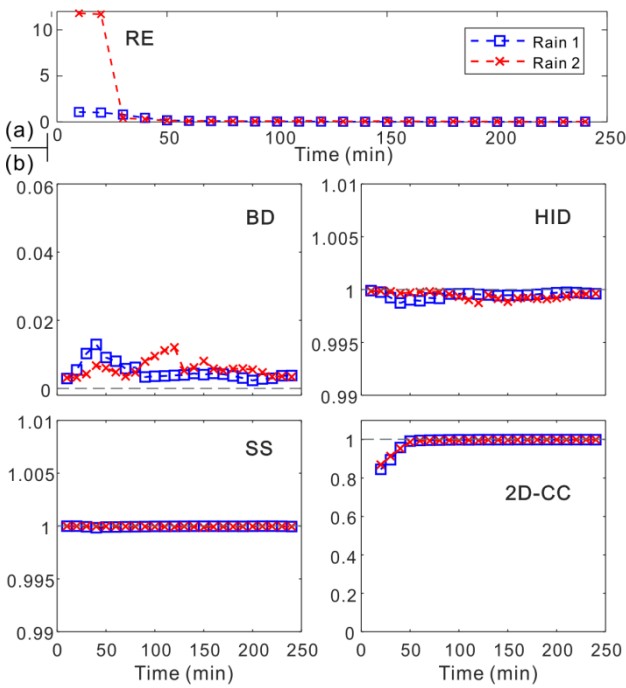

**Figure 17: (a) Relative error, and (b) Degree of similarity (BD, HID, SS and 2D-CC) of the flood predictions of testing rainfalls at different time steps**

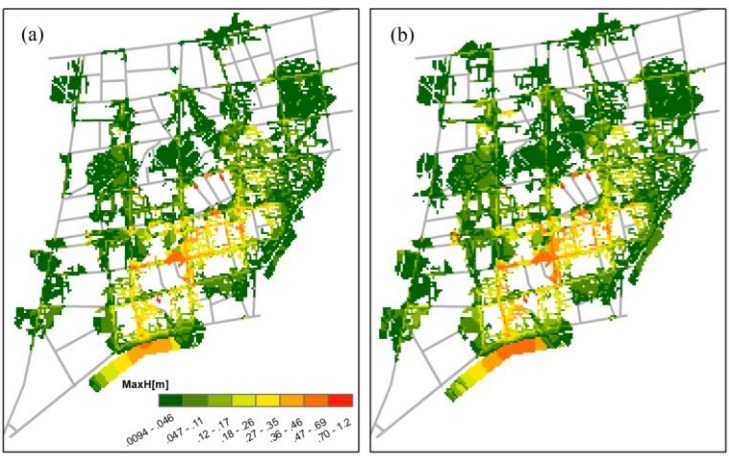

**Figure 18: Sample comparison of flood map of (a) ground truth and (b) model prediction in the second case study under a 50-year event.**

**Table 1: The performance indicators of the tested rainfalls in the second case study.**

| Rainfall events | Performance indicators | | | |
|:---:|:---:|:---:|:---:|:---:|
| | BD | HID | SS | 2D-CC |
| A | 0.003167 | 0.999400 | 0.999810 | 0.997707 |
| B | 0.003961 | 0.999227 | 0.999950 | 0.999361 |
| C | 0.010744 | 0.996130 | 0.997472 | 0.929869 |
| D | 0.003279 | 0.999480 | 0.999982 | 0.999637 |
| E | 0.009604 | 0.996510 | 0.997349 | 0.927005 |
| F | 0.003337 | 0.999381 | 0.999960 | 0.999301 |

Lastly, the proposed LSTM model was compared with the two baseline models (i.e., ANN and CNN) in Fig. 19. Our model outperformed the baseline models in terms of evaluation indicators on both the relative error and the degree of similarity. This confirmed the excellent performance of LSTM in flood predictions on water depth and spatial distribution. The ANN performed poorly in predicting water depths and there were a large number of cells associated with large errors. Regarding the BD, HID and SS, the CNN was the least ideal in predicting the spatial distributions. One possible reason could be that the convolution operation of CNN filtered part of the feature information of flood distribution. Note that the ANN's prediction based on the 2D-CC indicator was worst. This could be due to that the fully connected network structure of ANN was prone to overfitting, and may also be interfered by some redundant information. Furthermore, a sample illustration of the predicted flood maps by the three types of models is shown in Fig. 20. It is clear that our proposed model was more competitive in flood predictions than the other two classical methods.

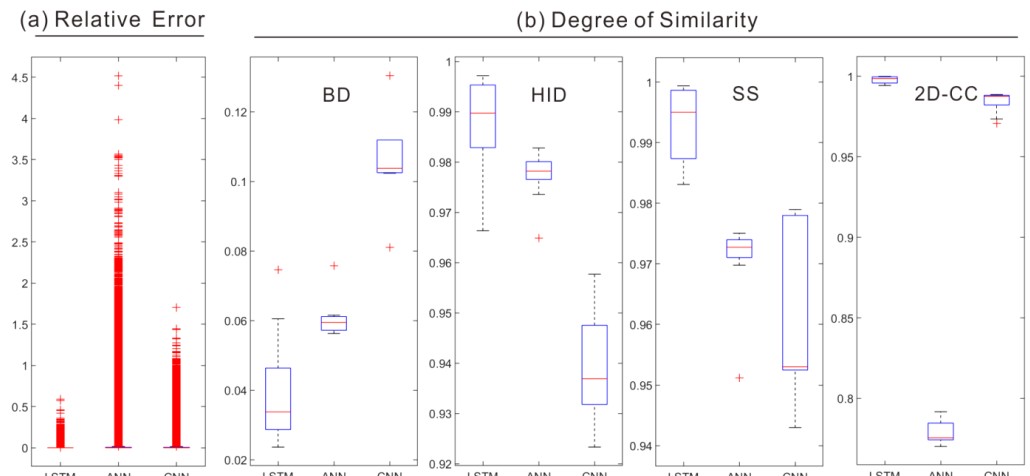

**Figure 19: (a) the mean relative error and (b) degree of similarity indicators of the proposed LSTM and two baseline models (ANN and CNN), respectively.**

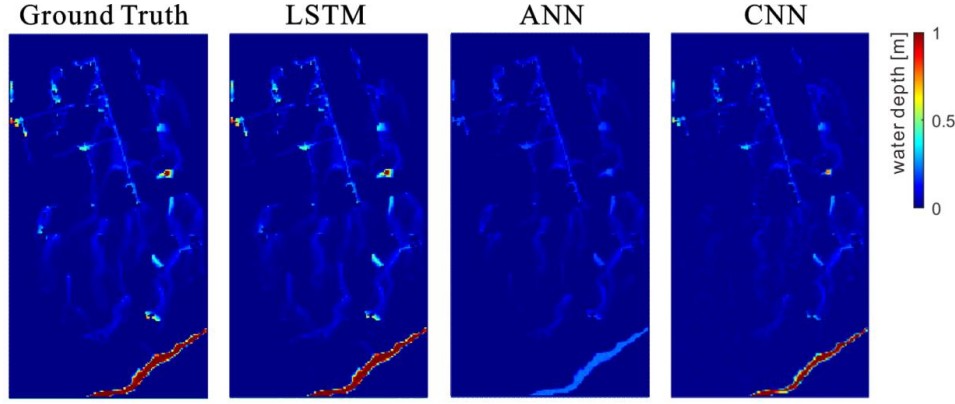

**Figure 20: A sample comparison of flood inundation maps of ground truth, LSTM, ANN and CNN models under an 85-year rainfall event.**

**4 Conclusions**

A rapid, accurate and dynamic flood prediction tool is of great significance to urban water management to protect people, social assets and environment from flood hazards. This study proposed a DL technique-based data-driven flood prediction approach, employing an integration of the LSTM technique, Bayesian optimization, and transfer learning approach. Results clearly showed that model could accurately produce both the maximum water depths, and the time series flood maps for various

hyetograph inputs with much lower computation costs. Such types of predictions on dynamic changes in both temporal and spatial scales are of great interest. By exploring the role of Bayesian Optimization algorithm in the LSTM network, the best design scheme of the network was determined. The results of our testing site showed that the LSTM could quickly adapt to the prediction task in the new site, and the transferred LSTM performed accurate flood predictions. The transfer learning method required less time,

lower resource cost, and delivered better real-time performance, especially when dealing with large-scale site information.

The predicted flood maps were 19,585 times faster than the hydrodynamic model. The achieved mean relative error in water depths is 9.5% and the degree of similarity of flood maps was very high. Specifically, in a best case, the difference between the ground truth and model prediction was only 0.76%

and the spatial patterns of the two types of maps were almost identical. Meanwhile, the transfer learning technology has greatly improved the compatibility and generalization ability of the proposed method. The superior model performance was further validated by comparing with the two baseline models. In conclusion, the accuracy and efficiency of the proposed method is satisfying.

We acknowledge some limitations in this study and discuss directions of future work. First of all,

the current training and testing data were obtained from hydrodynamic modelling due to a lack of detailed field site data. In future work, we consider adopting image capture techniques for data supplement, such as DL techniques for automated detection, acquisition and evaluation of water depths from camera images. In doing so, there will be more real case/field survey dataset for model training and testing. Meanwhile, the data augmentation is useful in enhancing the quantity and quality of input data,

which will be tested in future investigations.

Despite the limitations, this work with its advances can well contribute to a better understanding of the deep learning techniques for urban flood mapping. The proposed methodology predicts temporal and spatial water depths with only rainfall inputs, without further requirements of e.g., local terrains and

geographical conditions. The approach can be easily adjusted or adopted for other types of applications

495 in water management field. In summary, the method proposed represents a compromise solution that

takes into account prediction efficiency, accuracy, and adaptability. More importantly, the proposed

method can potentially replace and/or complement the conventional detailed hydrodynamic model for

urban flood assessment and management, particularly in applications of real time control, optimization

and emergency design and plan.

500 **Data Availability**

The dataset that support the findings of this study are available from the corresponding author upon

reasonable request.

**Author contributions**

QZ conceived the idea and acquired the project and financial support. QZ and ST designed the study, and

505 conducted all the experiments and analyzed the results. ZS, XL and JF collected and preprocessed the

data. QZ wrote the first draft of the manuscript with contributions from ST and ZS. ZS, XL, JF and GC

provided feedback on results and edited the manuscript. JZ and ZL provided the data and assisted with

analysis on the second case study and further improved the manuscript.

**Competing interests**

510 The contact author has declared that neither they nor their co-authors have any competing interests.

**Disclaimer**

Publisher's note: Copernicus Publications remains neutral with regard to jurisdictional claims in

published maps and institutional affiliations.

**Acknowledgements**

515 This research was funded by the Youth Promotion Program of the Natural Science Foundation of

Guangdong Province, China (Grant No. 2023A1515030126), National Natural Science Foundation of

China (Grant No. 51809049), the National College Students Innovation and Entrepreneurship Training

Program (Grant No. 202111845038).

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
