# Peer review of "A deep learning technique-based data-driven model for accurate and rapid flood predictions in temporal and spatial dimensions"

_Hydrology and Earth System Sciences, 2021_

## Author Response (AR1)

Paper ID: hess-2021-596 (A deep learning technique-based data-driven model for accurate and rapid flood prediction)

Dear Dr. Dimitri Solomatine,

We greatly appreciate the editor and reviewers for the constructive comments to improve the manuscript. We have addressed all the reviewers' comments in the revised version and the point-by-point responses are provided in the following Response Letters to each reviewer. Meanwhile, the detailed changes have been highlighted in the Annotated Version. It can be seen that the revised literature review, the improved Methodology, the added dataset and case study, and the relevant results and discussion have all been clearly presented to draw out the novelty and contribution of this study.

We sincerely hope you and the reviewers will find the revised version much more comprehensive and robust. All the authors have reviewed the manuscript and agreed to the submission of the manuscript. We look forward to hearing from you.

Thank you for your time and efforts on our manuscript again.

Sincerely Yours,
Dr. Qianqian Zhou
On behalf of all authors
School of Civil and Transportation Engineering, Guangdong University of Technology, No. 100 Waihuan Xi Road, Guangzhou 510006, China
E-mail: qiaz@foxmail.com

Paper ID: hess-2021-596 (A deep learning technique-based data-driven model for accurate and rapid flood prediction)

**Response to Reviewer 1**

Comment 1

"*I find the topic of the submitted manuscript very interesting and also within the scope of HESS. Rapid flood prediction mapping in urban areas has a few challenges when compared with flood modelling purposes (e.g., fluvial flood mapping). The authors identify these challenges and propose a data-driven flood model based on LSTM networks and Bayesian optimisation.*"

**Response:** We greatly thank the helpful and encouraging comments from the reviewer and the acknowledgment on the contribution and value of our work. We have fully addressed each of the reviewer's comments in the revised version and the paper has been greatly improved thanks to the reviewer.

Comment 2

*"The optimisation part is interesting, but the implementation and justification of the LSTM flood model lacks, in my opinion, novelty. One of the arguments of the authors for using LSTM to predict flooding is its suitability to predict time series, i.e. to include the time dimension in the flood prediction mapping. However, they fail to do so as they predict only the maximum water depth maps. This has been presented in previous recent studies (that are correctly acknowledged by the authors); so, what are the novel aspects of this study? Simple a different network architecture? I believe this is not for a scientific contribution that aims to contribute to the advance of the (applied) science."*

**Response:** We agree with the reviewer and have greatly enhanced the methodology and related descriptions and results. To improve the novelty of the paper, in the revised version we have revealed the advantages of the LSTM network on flood time series predictions. Specifically, the following contents have been incorporated:

> **(1) Create an additional dataset for flood time series.** To research the prediction performance of LSTM on flood time series, an additional dataset including 11 rainfall events and resulted flood time series has been included. The flooded water depths were recorded

every ten minutes for the entire case study under each rainfall. Among that, nine rainfalls were used for training and the other two rains for testing.

**(2) Explore the LSTM network for time series predictions**. With the revised methodology and dataset, the tested results showed that our model can well predict flood variations at different time steps. Figure 1 below shows the sample comparisons between ground truths and model predictions of flood maps in time dimension. In visual quality, the predicted flood maps were in good agreement with ground truths at all time steps. The overall prediction effects (based on the relative error) and the evaluation indicators on the degree of similarity were summarized in Figure 2. These indicators further validated the model performance, which was also satisfying in predicting flood time series.

[Figure]

Figure 1: Sample comparisons of flood maps between ground truths and model predictions at different time steps in the entire case study.

[Figure]

Figure 2: (a) Relative error and (b) Degree of similarity (BD, HID, SS and 2D-CC) of flood predictions at different time steps. Specifically, the closer the BD is to 0, and the HID, SS, 2D-CC are to 1, the better the prediction performance.

Comment 3

*"One of the main challenges about data-driven models, in general, is the capability of the models to generalize to different case studies or contexts. This aspect is not investigated nor discussed in the manuscript - it is only briefly mentioned, and for the 1st time, in the conclusions section. Since terrain elevation is not part of the input data set, it seems that the proposed model is not at all generalisable to other cases."*

**Response:** We agree with the reviewer that one of the main challenges of data-driven models is their compatibility. In the field of machine learning, the transfer learning (TL) technology can

significantly improve the application field of intelligent algorithms. In this paper, **we used transfer learning technology to transfer the LSTM network obtained from the current site (Site A) to another site (Site B), so as to expand the compatibility and generalization ability of our method.**

The TL is a deep learning method to transfer knowledge from one domain (source domain) to another domain (target domain), see Figure 3. Through the training of a source model (pre-trained network) using the source data (Site A), the pre-trained network can gain a strong ability of feature extraction. Subsequently, with fine-tuning (transfer learning) the new data (Site B), the pre-trained network can quickly adapt to the new site under different scenarios. With this method, a lot of training time can be saved for the target domain (the new site), and better training effects can be achieved, especially when there are limited training samples in the target domain.

[Figure]

Figure 3: Transfer learning technology.

In the revised version, we have included a new case study to test the performance of our method with the TL technology. The main landuse, drainage system and surface elevation are shown in Figure 4. The tested results (Figure 5) showed that with TL the proposed model was applicable and generalisable to other cases and **the predicted flood maps were consistent and similar to the ground truths.** Specifically, Table 1 shows the achieved performance indicators of all tested rainfall events. The obtained BD is close to 0, and HID, SS and 2D-CC are close to 1, which means the model predictions are highly similar to the ground truth results. **This further proves that the flood prediction of the new site can be realized through the TL technology**.

[Figure]

Figure 4: (a) landuse and (b) drainage network and DEM of the new case study.

[Figure]

Figure 5: Sample comparison of flood maps of (a) ground truth and (b) model prediction in the new case study under a 50-yr event.

Table 1: The performance indicators of tested rainfalls in the new case study.

| Rainfall events | Performance indicators | | | |
|---|---|---|---|---|
| | BD | HID | SS | 2D-CC |
| A | 0.003167 | 0.9994 | 0.99981 | 0.997707 |
| B | 0.003961 | 0.999227 | 0.99995 | 0.999361 |
| C | 0.010744 | 0.99613 | 0.997472 | 0.929869 |
| D | 0.003279 | 0.99948 | 0.999982 | 0.999637 |
| E | 0.009604 | 0.99651 | 0.997349 | 0.927005 |
| F | 0.003337 | 0.999381 | 0.99996 | 0.999301 |

Comment 4

"*The limitations of the model and study presented at the end of the Conclusions section are very similar to those of other previous studies. If the authors are aware of these limitations from previous studies, I would expect them to try to address at least some of the previous studies limitations to improve the knowledge.*"

**Response:** Thanks for the suggestions. We have improved the methodology and better explained the novelty of the revised paper. We have also better present the related literature and revised the Conclusions section. To summarize, the paper has been improved in three aspects following the reviewer's comments.

(1) Comment 2: enhance the model capability in **predicting flood time series,** highlighting the prediction effects of the LSTM network in time dimension

(2) Comment 3: the adoption of **transfer learning technology** to improve the compatibility and generalization ability of the proposed method

(3) Comment 5: explore the role of **Bayesian Optimization Algorithm** in the LSTM network, and the determination of best design scheme of the network

Comment 5

"*As mentioned above, I think the part of the optimisation could be better explored in the manuscript. Perhaps this could be the novel contribution of the study?*"

**Response:** Thanks for the suggestions. In the revised version, we have added more methodology descriptions and parameterization analysis of the Bayesian optimization. Specifically, it included the following contents:

(1) **Clarify more optimization process and details in the methodology section,** see the added workflow of the Bayesian optimization process in Figure 6.

The optimization process is implemented as follows: (1) A group of hyper-parameter combination $x_0$ (e.g., maxEpochs, learning rates) is randomly selected within the value ranges of the hyper-parameters (shown in Section 2.3.2); (2) The $x_0$ is input to the network for training to obtain the corresponding objective function $f(x_0)$; (3) The probability distribution of $f(x)$ corresponding to $x$ is calculated and predicted through the Gaussian process using all the inputs $(x, f(x))$; (4) The optimal $x$ is determined by the acquisition function in the probability distribution; (5) The $x$ obtained from Step (4) is taken as the

hyper-parameter combination of the network to train, and calculate the objective function $f(x)$; (6) Before reaching the maximum iteration number, the $(x, f(x))$ obtained in Step (5) will be used as the input of the Gaussian process to continuously update the probability model to obtain a new $(x, f(x))$. Once the maximum iteration number is reached, the $x$ corresponding to the minimum value of $f(x)$ is taken as the optimal hyper-parameter combination $x^*$.

[Figure]

Figure 6: The Bayesian optimization process.

(2) In the **Result section**, we **analyzed the influence of network parameters** (i.e., network depth, Epoch, MinBatchSize and initial learning rate) **on the prediction results** (see Figure 7).

The results show that: (1) There are large errors when the values of MaxEpochs (i.e., maximum number of epochs) are set too low. Increasing the number of training epochs can avoid adverse events. (2) The MiniBatchSize has little influence on the prediction results, but it is not appropriate to take too large or small values. In this case, the MiniBatchSize of 20-70 can ensure an ideal prediction effect. (3) It is recommended to set a low learning rate.

When the value is low, the achieved error is small and close to 0. (4) A deeper network layer can obtain a smaller prediction error. With the parameterization analysis, the best design scheme of the LSTM can be determined through the optimization.

[Figure]

Figure 7: Influence of four types of network parameters on model prediction performance (i.e., relative error).

Comment 6

"*The manuscript is, in general, well written, making it easy to read.*"

**Response:** We thank the reviewer for the positive feedback and enlightening advices.

**Specific comments:**

1. "*Line 43: this is valid also for physically-based models. Rephrase?*"

**Response:** Comments incorporated in the revised version.

2. "*Line 113 & 115: why different models Mike Urban & Flood vs Mike 21? Please provide justification or mention only the model used.*"

**Response:** We agree and have clarified in the revised version that 'Mike Urban' was used as the physical model.

3. *"Line 145: can "... for the long-term memory of data" be better described?"*

**Response:** Yes, we have replaced "the long-term memory of data" with "the time-varying data".

4. *"Line 149: unclear sentence. It seems that something is missing."*

**Response:** We agree and have added a new sentence to better explain the information: "It is a priori probability that is used as the basis for selecting the parameter combination in the next iteration".

5. *"Lines 279 - 282: this can't be seen in the plots. The colour scale does not have units. How does the colour scale relate to the yy axis?"*

**Response:** We have revised the original Figure 9 and Figure 10, and the related legends, captions and texts to better clarify the information and discuss the results. The units of color scales in Figure 9 and 10 are meter (i.e., water depth) and relative error (-), respectively.

[Figure]

Figure 9: Sample comparison of flood maps between ground truth and model prediction in the best case scenario.

[Figure]

Figure 10: Relative error of (a) best (i.e., minimum), (b) mean, and (c) worst (i.e., maximum) case scenarios, respectively. (d) summarizes the relative error data in boxplots for the three types of scenarios.

*6.    "Line 306: the worst and best cases are also interesting to be analyzed and discussed."*

**Response:** We agree and have added more data and discussions on the worst and best case scenarios. Figure 10 has been revised to better present the information. It is shown in Figure 10d that in all cases, the mean errors were below 1%. The errors were much higher in the worst case, where there were a small number of cells associated with relative errors greater than 20%. These high-error cells were mainly located in/near the water bodies.

Paper ID: hess-2021-596 (A deep learning technique-based data-driven model for accurate and rapid flood prediction)

**Response to Reviewer 2**

Comment 1

"*The author used the rainfall as the input to generate simulated flood inundation maps. The paper is well organized, and the LSTM model and Bayesian optimization method appears to be correct and effective. However, there are three major issues. First, the summary of the earlier work needed improvement. There are many related papers using data-driven approach to generate flood maps, however the authors do not include them in the introduction. Second, a research may be regarded as a novel study if it resolves a problem or constraint in earlier studies. However, the LSTM is a new neural network layer that can perform better than ANN or linear regression models, and this manuscript does not appear to have demonstrated its novelty. Lastly, this manuscript lacked baseline models and results, which prevented me from knowing how much better the LSTM model is than a simple average baseline, linear regression model, or an ANN model.*"

**Response:** We greatly thank the reviewer for the valuable comments to improve the paper. We have significantly revised the paper following the suggestions and better explained the novelty of the work. Overall, the paper has been improved in the following aspects:

(1) We have **improved the summary of the previous work** and added more related papers and suggested references to better explain the novelty of the proposed methodology.

(2) We have **enhanced the prediction effects of the LSTM network in predicting not only the maximum water depths, but also flood time series**, which was a constraint/has not been reported in the previous literature. The proposed method enabled the flood prediction in time dimension, which is a new contribution to the field.

(3) The revised paper adopted the **transfer learning technology to improve the compatibility and generalization ability** of the proposed model. New results showed the model can be well applied to other case studies. The practical application prospect of the proposed method is enhanced.

(4) We have **analyzed the role of Bayesian Optimization Algorithm in the LSTM network**, and the determination of best design scheme of the network. This has not been explored in the previous literature.

(5) Following the reviewer's suggestions, **we have added two flood prediction algorithms**

**(ANN and CNN) as the baseline models to confirm the effectiveness** of the proposed method. New results show that the LSTM is more competitive than the other two algorithms and performed better in terms of evaluation indicators.

Comment 2

*"Method section 2.2. If my understanding is correct, all the flood maps are simulated by your physically-based model. Thus, your are developing a deep learning model as a surrogate model of your Mike series models. Such studies have been studied in the past several years using Deep Learning models (see the following papers). If the main difference between your study and theirs is the use of LSTM other than a fully connect layer, this is not novel enough.*

*Berkhahn, S., Fuchs, L., & Neuweiler, I. (2019). An ensemble neural network model for real-time prediction of urban floods. Journal of hydrology, 575, 743-754; Lin, Q., Leandro, J., Wu, W., Bhola, P., & Disse, M. (2020). Prediction of maximum flood inundation extents with resilient backpropagation neural network: case study of Kulmbach. Frontiers in Earth Science, 8, 332."*

**Response:** First of all, sorry for insufficiently addressing the novelty of the proposed method in the original version. As responded to Comment 1 that we have significantly revised the paper (including the introduction, methodology and related results) to better explain the contribution of our work. One of the novelties is to predict flood maps in time dimensions (Figure 1 below). Also the Bayesian optimization approach has significantly improved the accuracy of prediction.

[Figure]

Figure 1: Sample comparisons of flood maps between ground truths and model predictions at different time steps in the entire case study.

Secondly, this study is aimed to develop a flood prediction model that is applicable to various types of case studies, not just as a surrogate model of the physical model. The physical model was used to provide training samples for our model to learn the flood feature extraction ability. In the revised version we used the transfer learning (TL) technology to transfer the LSTM network obtained from the current site (Site A) to another site (Site B), to expand the compatibility and generalization ability of our method. A new case study (Figure 2) was used to test the performance of the method. Results (Figure 3) showed that with TL the proposed model can be well adapted to other cases and the predicted flood maps were consistent to the ground truths. Specifically, the model performance indicators in the new case are shown in Table 1. It is shown that the predicted results are very satisfying (BD is close to 0, HID, SS and 2D-CC are close to 1), which proves that flood disaster prediction of the new site can be realized through the TL technology.

[Figure]

Figure 2: (a) main landuse and (b) drainage network and DEM of new case study.

[Figure]

Figure 3: Sample comparison of flood maps of (a) ground truth and (b) model prediction in the new case study under a 50-yr event.

Table 1: The performance indicators of all tested rainfall events in the new case study.

| Rainfall events | Performance indicators | | | |
|---|---|---|---|---|
| | BD | HID | SS | 2D-CC |
| A | 0.003167 | 0.9994 | 0.99981 | 0.997707 |
| B | 0.003961 | 0.999227 | 0.99995 | 0.999361 |
| C | 0.010744 | 0.99613 | 0.997472 | 0.929869 |
| D | 0.003279 | 0.99948 | 0.999982 | 0.999637 |
| E | 0.009604 | 0.99651 | 0.997349 | 0.927005 |
| F | 0.003337 | 0.999381 | 0.99996 | 0.999301 |

Comment 3

*"What is the color in Figure 6a represents? Can you provide more details about this figure? It seems like the increase of the number of optimizations does not decrease the error much."*

**Response:** We thank this comment to improve the presentation of the optimization results. The color of the bubble chart in Figure 6a represents the size of the error value. We have revised the chart to better demonstrate the variations of the relative errors along with the iterations.

[Figure]

Figure 6: The mean relative errors along with the Bayesian optimization process.

Comment 4

*"Are your figures 9 and 10 captions correct? And, is your legend correct for Figure 10? The base color Cyan should represent 0 on your Y-axis, but the legend shows it is 0.5 relative error."*

**Response:** We thank the reviewer for pointing out the problems. Both figure captions were typos and we have corrected them in the revised version. The data of Figure 10 was correct, but the base color was misleading in the previous version because we adopted the 'shading interpolation' function to visualize

the figure. We have revised Figure 10 to better present the information and added a subplot to summarize the data for different case scenarios.

[Figure]

Figure 10: Relative error of (a) best (i.e., minimum), (b) mean, and (c) worst (i.e., maximum) case scenarios, respectively. (d) summarizes the data in boxplots for the three types of scenarios.

Comment 5

*"Can you provide results from several baseline models to justify your model performance is good? Some sample baselines could be: 1, models such as ANN as Berkhahn, S., Fuchs, L., & Neuweiler, I. (2019) did (deep learning model using only FC layers other than LSTM). 2, a Lasso or Ridge Regression (or machine learning models) for each point with the overall rainfall as input, water depth as output. 3, an average/median flood map of the training dataset (a.k.a. simple average, see the link below). Without these baselines, your results in Figure 8a and 8b cannot prove much -- we know your model is good, but we don't know how good your model comparing to other simple linear models or simple average of training sets.*

*https://otexts.com/fpp2/simple-methods.htm"*

**Response:** Following your suggestions, we have investigated two additional flood models (ANN and CNN) to justify the performance of our model. We will explain more details on the ANN (a fully

connected network (Back-Propagation neural network)) and CNN (including two convolution layers, one pooling layer and a fully connection layer) models in the revised literature review and Methodology section.

The new results (Figure 4) showed that the LSTM model outperformed the two baseline models in terms of evaluation indicators on both the relative error and the degree of similarity. This confirms the excellent performance of LSTM in flood prediction on water depth and spatial distribution. The ANN performed poorly in predicting water depths and there were a large number of cells associated with large errors. Regarding the BD, HID and SS, the CNN was the least ideal in predicting the spatial distributions. One possible reason could be that the convolution operation of CNN filtered part of the feature information of flood distribution. Note that the ANN's prediction based on the 2D-CC indicator was worst. This could be due to that the fully connected network structure of ANN was prone to overfitting, and may also be interfered by some redundant information.

[Figure]

Figure 4: (a) the mean relative error and (b) degree of similarity indicators of the proposed LSTM and two baseline models (ANN and CNN), respectively.

Furthermore, a sample illustration of the predicted flood maps by the three types of models is shown in Figure 5. It is clear that in visual quality our proposed model is also more competitive in flood predictions than the other two methods.

[Figure]

Figure 5: A sample comparison of flood inundation maps of ground truth, LSTM, ANN and CNN models under an 85-year rainfall event.

---

## Author Response (AR2)

Paper ID: hess-2021-596 (A deep learning technique-based data-driven model for accurate and rapid flood predictions in temporal and spatial dimensions)

Dear Dr. Dimitri Solomatine,

We greatly appreciate the additional comments to improve the revised manuscript. We have addressed all the reviewers' comments in the resubmitted version and the point-by-point responses are provided in the Response Letters to each reviewer. The detailed changes have been highlighted in the Annotated Version. We sincerely hope that you and the reviewers will find the revised version more scientifically clear and robust. We look forward to hearing from you soon.

Thank you for your time and efforts on our manuscript again.

Sincerely Yours,
Dr. Qianqian Zhou
On behalf of all authors
School of Civil and Transportation Engineering, Guangdong University of Technology, No. 100 Waihuan Xi Road, Guangzhou 510006, China
E-mail: qiaz@foxmail.com

Paper ID: hess-2021-596 (A deep learning technique-based data-driven model for accurate and rapid flood predictions in temporal and spatial dimensions)

**Response to Editor**

Comment 1

 *"I have now finally both reviews - but based on the comments of both referees I have to kindly ask you for an additional revision. This is a "minor revision". I would like to ask to give special attention to the comments made by Referee 1. I am citing:*

*"(1)...after reading the manuscript again, it is still unclear what are the input data for the LSTM flood prediction model presented in the manuscript. More specifically, I could not find in the document that DEM is one of the input data along with the rainfall event (see Figure 3). This seems to me a little bit odd, as far as I can understand without the DEM, i.e. without providing the terrain features as input data, the proposed LSTM flood prediction model would struggle, if not impossible, to be applicable to case studies (catchments) other than the one used for training. This is a STRONG problem the authors do not clearly explain. In order to make the scientific development presented meaningful and trustful, this aspect needs, in my opinion, to be clarified.*

*(2) As the authors mention, the key for successful data-driven simulations is the number and variety of the data used for training the LSTM (and other data-driven models). In the manuscript, very little information is provided about the data set used for training. How many simulations? how different were the simulations? ...*

*Based on the two points raised above, which for me are critical, I suggest the authors to attempt an additional revision.""*

**Response:** Thanks for the comments to better clarify the methodology. In the revised version, we have provided additional details to explain 1) LSTM methodology and input data requirement, 2) the number and variety of input data. Both comments have been properly addressed, please find more details in the Response Letter to Referee 1.

Comment 2

*"Further, this referee points out that it is important to address these two comments in a meaningful way, especially the first comment. I agree with this opinion. Please share your*

*thoughts and provide explanations about the possibilities of using the trained data-driven model(s) for other catchments.”*

**Response:** We have better explained the use of transfer learning technique in adapting the trained model for other catchments. The method proposed in this paper is a compromise between prediction accuracy, computational efficiency and adaptability, which can achieve excellent flood prediction results. More details can be found in the Response Letter to Referee 1.

Paper ID: hess-2021-596 (A deep learning technique-based data-driven model for accurate and rapid flood predictions in temporal and spatial dimensions)

**Response to Reviewer 1**

Comment 1

"*I acknowledge the substantial revisions the authors conducted in the original version of the manuscript. Most of the reviewers' comments were adequately addressed.*"

**Response:** We greatly thank your helpful suggestions to improve the paper and acknowledgment of our efforts in the first revision.

Comment 2

"*However, after reading the manuscript again, it is still unclear what are the input data for the LSTM flood prediction model presented in the manuscript. More specifically, I could not find in the document that DEM is one of the input data along with the rainfall event (see Figure 3). This seems to me a little bit odd, as far as I can understand without the DEM, i.e. the terrain features, the proposed LSTM flood prediction model would struggle, if not impossible, to be applicable to other case studies (catchments) other than the one used for training.*"

**Response:** We agree that the terrain features are a key factor in implementing flood prediction tasks. Besides, other catchment hydrological properties (e.g., land use, area and imperviousness) and network parameters (pipeline distribution and capacity) also have essential impacts on flood conditions. When designing the method, we actually considered the impacts of all influencing factors in the network training, but without specifying/demanding them in the model input. Instead, we used the mapping relationships between rainfall and flood depth to reflect the impact of these factors on flood through the established LSTM, i.e., regression model. We will clarify this by the following points:

(1) The site information contains a vast amount of data. Although computer technology has made significant progress recently, deep learning algorithms still face significant challenges in processing high-dimensional data. Such data can significantly decrease the prediction efficiency of the algorithm, particularly when applied to large-scale areas. This will affect/harm the real-time performance of the prediction model.

(2) Neural network technology involves learning and creating a mapping relationship between input and output data. In this study, the input data is rainfall intensity, and the output is flood depth. As mentioned previously, the depth of rainfall is influenced by e.g., terrain features, catchment hydrological factors, network parameters. The function of neural network is to establish the relationship between the input, these implicit influence conditions and the output. The results of this study demonstrate that LSTM can accurately establish the above relationships. As a result, the method proposed took into account the impact of terrain features and other influencing factors in the network model.

(3) We understand the reviewer's concern on the applicability of the model, as it appears that the model established is only applicable to the investigated site. The problem can be solved by using transfer learning technology to implement flood prediction for new sites. Data of the new site is used to fine-tune the LSTM model obtained from the old site (more details can be found in the Methodology－2.3.3 Transfer learning (TL)). The results of our testing site showed that the LSTM could quickly adapt to the prediction task in the new site, and the transferred LSTM performed accurate flood predictions. Especially when dealing with large-scale site information, the transfer learning method requires less time, lower resource cost, and delivers better real-time performance.

To summarize, this paper proposes a flood prediction model based on three key aspects: 1) mitigating the impacts of high-dimensional site feature data on the efficiency of the prediction model; 2) establishing an accurate prediction model using only rainfall intensity and flood depth data, while leveraging the nonlinear mapping characteristics of input, site features and output based on the LSTM network; 3) employing transfer learning technology to achieve the prediction task in new sites. As such, the method proposed in this paper represents a compromise solution that takes into account prediction efficiency, accuracy, and adaptability.

Comment 3

*"As the authors mention, the key for successful data-driven simulations is the number and variety of the data used for training the LSTM (and other data-driven models). In the manuscript, very little information is provided about the data set used for training. How many simulations? how different were the simulations? ..."*

**Response:** We have provided more details on the input data. Specifically, two types of datasets were established: (1) maxH dataset, i.e., the maximum flood depth: there were in total 90 rainfall events adopted, with return periods ranging from 1 to 100 years and rainfall duration of 2, 4 or 6 hours, respectively. Meanwhile, three types of rain peak position coefficients were tested, including 0.3, 0.5 and 0.7. That means there were 90 simulations from the physical model. The details on the return period, rainfall duration and peak position of the input rainfall events are provided in the attached Figure 1; (2) time series dataset: we adopted 11 rainfall events and recorded the flooded water depths every 10 minutes for the entire case study under each rainfall. As for the training process of the network, 90% of the rain intensities were used as training samples, and 10% as testing samples. In addition, 170 Bayesian optimizations were performed to identify the optimal network structure, which achieved the best prediction error after 385 training iterations.

[Figure]

Figure 1: Details of simulated rainfall events.

Paper ID: hess-2021-596 (A deep learning technique-based data-driven model for accurate and rapid flood predictions in temporal and spatial dimensions)

**Response to Reviewer 3**

Comment 1

"*LSTM is well-documented in literature in dealing with time series predictions. Obviously, this work's novelty is that the authors applied LSTM to a spatial and temporal forecasting task. My major concern is that the authors failed to elaborate well enough on how they transformed the spatial forecasting tasks so that LSTM can handle them. For instance, does the model output grided predictions, such as water depths, for the 27,183 grid points in the study area all at once? If so, how? Or, does it predict water depth for those grid points one at a time? Or, does it only predict at a few locations and then some extrapolation algorithms were used to obtain the flood map for the entire area?*"

**Response:** Thanks for the comments. In this paper, the output of LSTM is the water depth of all grid points all at once. A regression algorithm is used for the LSTM model. Specifically, the input rainfall intensity is processed through multiple LSTM layers and activation layers, and finally, a regression layer outputs the water depth of the 27,183 grid points for the entire area. A regression case (https://doi.org/10.1016/j.ijfatigue.2022.106812) has demonstrated the superior performance of the regression layer. In other words, the process is akin to a fitting process, in which different rainfall intensities are matched nonlinearly to the water depth of grid points. The number of output grids can be set during LSTM modeling so that the output grid can be specified for different sites.

Comment 2

"*Lines 100-106: Normally, we don't put results in the introduction section. As the opening sentence of this paragraph states, you should focus only on why you are doing this, how you plan to do this (in brief), and how your work is going to close some of the gaps in the literature. You can keep most of your writings in this paragraph as they are and simply remove those results.*"

**Response:** We agree with the reviewer and have removed the relevant text.

Comment 3

*"Figures 1 and 2: Consider adding a scale bar for those two study areas."*

**Response:** Both figures have been revised as suggested.

Comment 4

*"Line 142: Maximum water depth about what? As you indicated in Line 372, there are 27,183 grid points on your map. Do you create a maximum water depth record for each of those grid points? Do you aggregate them following some algorithms? Or do you only utilize water depth at certain locations, such as those manholes? The same question is also here for your output. Does your LSTM output a 2D map directedly? Does it predict the flood depth for grid points one at a time? Does it only predict the depths for a few points then some interpolation methods were used to expand from the scattered predictions to the entire* study area?"

**Response:** Sorry for the confusion. As explained previously (Comment 1), the maximum water depth is for the entire area (i.e., 27,183 grid points). LSTM outputs a 2D map directly, which describes the water depth of all grid points all at once. We have better clarified this information in the revised version.

Comment 5

*"Figure 3: What is the unit of T here? I assume it should be [year] as indicated by the caption of Figure 3. In the hydro-science research field, some of the most widely used return periods include 2-, 10-, 50-, 100-, 200-, and 500-yr. Therefore, I was wondering about the case where T=90. You should consider explaining why there is such a case with a return period of 90 years somewhere between lines 140 and 144 where you introduce your data. This could be how you determined those return periods for each rainfall event if you manually compute those statistics or a brief introduction about how your reference source (if the rainfall data are obtained from any existing data sources) assigned those rainfall events with various return periods"*

**Response:** The reviewer is right that the unit of T is [year] and we have added the information in the revised caption. Moreover, we agree with the reviewer about the choice of the representative return periods, and have added more information on the rainfall events, and revised Figure 3 by providing a case with a return period of 100 years (instead of the 90 years).